# One-for-All: Bridge the Gap Between Heterogeneous Architectures in Knowledge Distillation

**Zhiwei Hao**[1,2]**, Jianyuan Guo**[3]**, Kai Han**[2]**, Yehui Tang**[2]**,**
**Han Hu**[1*]**, Yunhe Wang**[2*]**, Chang Xu**[3]

[1]School of information and Electronics, Beijing Institute of Technology.
[2]Huawei Noah's Ark Lab.
[3]School of Computer Science, Faculty of Engineering, The University of Sydney.

{haozhw, hhu}@bit.edu.cn, jguo5172@uni.sydney.edu.au,
{kai.han, yehui.tang, yunhe.wang}@huawei.com, c.xu@sydney.edu.au

## Abstract

Knowledge distillation (KD) has proven to be a highly effective approach for enhancing model performance through a teacher-student training scheme. However, most existing distillation methods are designed under the assumption that the teacher and student models belong to the same model family, particularly the hint-based approaches. By using centered kernel alignment (CKA) to compare the learned features between heterogeneous teacher and student models, we observe significant feature divergence. This divergence illustrates the ineffectiveness of previous hint-based methods in cross-architecture distillation. To tackle the challenge in distilling heterogeneous models, we propose a simple yet effective one-for-all KD framework called OFA-KD, which significantly improves the distillation performance between heterogeneous architectures. Specifically, we project intermediate features into an aligned latent space such as the logits space, where architecture-specific information is discarded. Additionally, we introduce an adaptive target enhancement scheme to prevent the student from being disturbed by irrelevant information. Extensive experiments with various architectures, including CNN, Transformer, and MLP, demonstrate the superiority of our OFA-KD framework in enabling distillation between heterogeneous architectures. Specifically, when equipped with our OFA-KD, the student models achieve notable performance improvements, with a maximum gain of 8.0% on the CIFAR-100 dataset and 0.7% on the ImageNet-1K dataset. PyTorch code and checkpoints can be found at https://github.com/Hao840/OFAKD.

## 1 Introduction

In recent years, knowledge distillation (KD) has emerged as a promising approach for training lightweight deep neural network models in computer vision tasks [1, 2, 3, 4]. The core idea behind KD is to train a compact student model to mimic the outputs, or soft labels, of a pretrained cumbersome teacher model. This method was initially introduced by Hinton *et al.*. [5], and since then, researchers have been actively developing more effective KD approaches. One notable improvement in KD methods is the incorporation of intermediate features as hint knowledge [6, 7, 8]. These approaches train the student to learn representations that closely resemble those generated by the teacher, and have achieved significant performance improvements.

---

[*]Corresponding author.

37th Conference on Neural Information Processing Systems (NeurIPS 2023).

Existing hint-based approaches have predominantly focused on distillation between teacher and student models with homogeneous architectures, *e.g.*, ResNet-34 *vs.*ResNet-18 in [9], and Swin-S *vs.*Swin-T in [8], leaving cross-architecture distillation relatively unexplored. However, there are practical scenarios where cross-architecture distillation becomes necessary. For example, when the objective is to improve the performance of a widely used architecture like ResNet50 [10], as our experiments will demonstrate, distilling knowledge from ViT-B [11] to ResNet50 can readily surpass the performance achieved by distilling knowledge from ResNet152. Cross-architecture distillation thus presents a viable alternative, providing more feasible options for teacher models. It may not always be possible to find a superior teacher model with a homogeneous architecture matching the student's architecture. Moreover, the emergence of new architectures [12, 13] further compounds this challenge, making it difficult to find readily available pre-trained teacher models of the same architecture online. This limitation necessitates the need for researchers to resort to pre-trained teachers of different architectures for knowledge distillation in order to enhance their own models.

The few attempts that have been made in this direction mainly involve using heterogeneous CNN models for knowledge transfer [14, 15, 16]. In a recent study, Touvron *et al*. [17] successfully trained a ViT student by using a CNN teacher and obtained promising results. These endeavors have sparked our interest in studying the question: whether it is possible to ***distill knowledge effectively across any architectures***, such as CNN [10, 18, 19], Transformer [20, 21, 11, 17, 22], and MLP [13, 23, 24].

In previous studies where teachers and students share similar architectures, their learned representations naturally exhibit similarity in the feature space. Simple similarity measurement functions such as mean square error (MSE) loss, are sufficient for information distillation [6]. However, in the case that the teacher and student architectures are heterogeneous, there is no guarantee of successful alignment of the learned features, due to the fact that features from heterogeneous models reside in different latent feature spaces, as shown in Figure 1. Therefore, directly matching these irrelevant features is less meaningful and even impedes the performance of the student model, as shown in the third row of Table 1, highlighting the difficulty in exploring cross-architecture distillation at intermediate layers.

To address above challenges, we propose a one-for-all KD (OFA-KD) framework for distillation between heterogeneous architectures, including CNNs, Transformers, and MLPs. Instead of directly distilling in the feature space, we transfer the mismatched representations into the aligned logits space by incorporating additional exit branches into the student model. By matching the outputs of these branches with that from teacher's classifier layer in logits space, which contains less architecture-specific information, cross-architecture distillation at intermediate layers becomes achievable. Furthermore, considering that heterogeneous models may learn distinct predictive distributions due to their different inductive biases, we propose a modified KD loss formulation to mitigate the potential impact of irrelevant information in the logits. This includes the introduction of an additional modulating parameter into the reformulated loss of the vanilla KD approach. The modified OFA loss adaptively enhances the target information based on the predictive confidence of the teacher, effectively reducing the influence of irrelevant information. To evaluate our proposed framework, we conduct experiments using CNN, Transformer, and MLP architectures. We consider all six possible combinations of these architectures. Our OFA-KD framework improves student model performance with gains of up to 8.0% on CIFAR-100 and 0.7% on ImageNet-1K datasets, respectively. Additionally, our ablation study demonstrates the effectiveness of the proposed framework.

## 2   Related works

Recently, significant advancements have been made in the design of model architectures for computer vision tasks. Here we briefly introduce two prominent architectures: Transformer and MLP.

**Vision Transformer.** Vaswani *et al*. [25] first proposed the transformer architecture for NLP tasks. Owing to the use of the attention mechanism, this architecture can capture long-term dependencies effectively and achieves remarkable performance. Inspired by its great success, researchers have made great efforts to design transformer-based models for CV tasks [26, 27, 28]. Dosovitskiy *et al*. [11] split an image into non-overlapped patches and projected the patches into embedding tokens. Then the tokens are processed by the transformer model like those in NLP. Their design achieves state-of-the-art performance and sparks the design of a series of following architectures [21, 29].

**MLP.** MLP has performed inferior to CNN in the computer vision field for a long time. To explore the potential of the MLP, Tolstikhin *et al.* [13] proposed MLP-Mixer, which is based exclusively on the MLP structure. MLP-Mixer takes embedding tokens of an image patch as input and mixes channel and spatial information at each layer alternately. This architecture performs comparably to the best CNN and ViT models. Touvron *et al.* [24] proposed another MLP architecture termed ResMLP.

The most advanced CNN, Transformer, and MLP achieve similar performance. However, these architectures hold distinct inductive biases, resulting in different representation learning preferences. Generally, there are striking differences between features learned by heterogeneous architectures.

**Knowledge distillation.** KD is an effective approach for compressing cumbersome models [30, 31, 32, 33, 34, 35], where a lightweight student is trained to mimic the output logits of a pretrained teacher. Such an idea was originally proposed by Bucila *et al.* [36] and improved by Hinton *et al.* [5]. Following works further improve logits-based KD via structural information [37, 38], model ensemble [39], or contrastive learning [40]. Recently, Touvron *et al.* [17] proposed a logits distillation method for training ViT students. Huang *et al.* [9] proposed to relax the KL divergence loss for distillation between teacher and student with a large capacity gap. Besides logits, some KD approaches also use intermediate features as hint knowledge. Romero *et al.* [6] first proposed the hint-based distillation approach. They adopted a convolutional layer to project student features into the space of the teacher feature size, and then constrained the projected feature to be like the teacher feature via mean square loss. Zagoruyko *et al.* [7] proposed to force the student to mimic attention maps. Yim *et al.* [41] adopted the flow of solution procedure matrix generated by the features as hint knowledge. There are also many other feature distillation approaches using various hint design [42, 43, 8]. Recently, applying KD to compress dataset has emerged as a popular topic [44, 45, 46].

Although existing hint-based distillation methods have achieved remarkable performance, they assume that student and teacher architectures are homogeneous. When the architectures are heterogeneous, existing approaches may fail because features of the student and the teacher are distinct. How to conduct hint-based distillation under this circumstance is still an open problem.

## 3 Method

### 3.1 Revisit feature distillation for heterogeneous architectures

**Discrepancy among architectures.** For a long time, CNN is the dominant architecture for CV tasks. Its inductive biases of locality and spatial invariance are strong priors for learning natural images, and restrict CNNs from learning global representations. Vision transformers [11, 21, 47] inherited from the NLP field are attention-based models. This architecture lacks inductive biases inherent to CNNs and thus can learn global representations easily. Moreover, inspired by the success of ViTs, pure MLP-based vision models have been proposed [13, 24], which are also low-priori models.

**Knowledge distillation.** KD trains a tiny student model to learn knowledge from a large pre-trained teacher model. Logits and features are the most common used types of knowledge in KD.

Logits depict the predictive distribution of a model. Logits-based distillation trains the student to mimic output logits of the teacher, which can be formulated as:

$$\mathcal{L}_{\text{KD}} = \lambda \mathbb{E}_{(\boldsymbol{x},y)\sim(\mathcal{X},\mathcal{Y})}[\mathcal{D}_{\text{CE}}(\boldsymbol{p^s}, y) + (1-\lambda)\mathcal{D}_{\text{KL}}(\boldsymbol{p^s}, \boldsymbol{p^t})], \tag{1}$$

where $(\mathcal{X}, \mathcal{Y})$ is the joint distribution of samples and class labels. $\boldsymbol{p^s}$ and $\boldsymbol{p^t}$ are predictions of the student model and the teacher model on sample $\boldsymbol{x}$, respectively. $\mathcal{D}_{\text{CE}}$ denotes the cross-entropy function. $\mathcal{D}_{\text{KL}}$ denotes the Kullback-Leibler divergence function. $\lambda$ is a hyperparameter controls the trade-off between one-hot label $y$ and soft label $\boldsymbol{p^t}$.

hint-based distillation adopts more fine-grained teacher activations at intermediate layers to train the student, which can be formulated as:

$$\mathcal{L}_{\text{FD}} = \mathbb{E}_{(\boldsymbol{x},y)\sim(\mathcal{X},\mathcal{Y})}[\textstyle\sum_i ||\mathbf{F}_i^T - \psi(\mathbf{F}_i^S)||_2], \tag{2}$$

where $\mathbf{F}^T$ and $\mathbf{F}^S$ are features of the teacher and the student at layer $i$, respectively. $\psi$ is a feature projector that maps student features to match dimension of teacher features [6]. Knowledge from logits and features are not opposites. In most cases, feature distillation approaches also adopt logits as an auxiliary supervision.

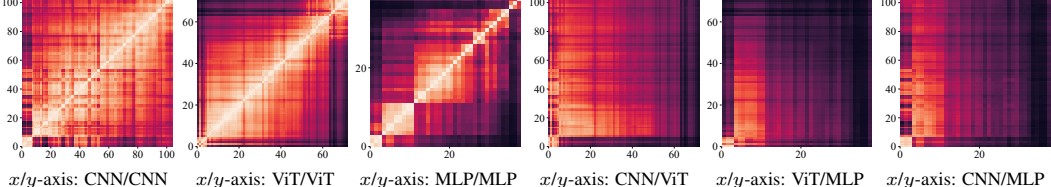

| $x/y$-axis: CNN/CNN | $x/y$-axis: ViT/ViT | $x/y$-axis: MLP/MLP | $x/y$-axis: CNN/ViT | $x/y$-axis: ViT/MLP | $x/y$-axis: CNN/MLP |

Figure 1: Similarity heatmap of intermediate features measured by CKA. We compare features from **MobileNetV2** (CNN), **ViT-Small** (Transformer) and **Mixer-B/16** (MLP model). The first three figures illustrate similarity between homogeneous models, and the last three illustrate feature similarity between heterogeneous models. Coordinate axes indicate the corresponding layer index.

**Challenges in heterogeneous feature distillation.** Existing feature distillation methods are designed under the assumption of teacher and student belonging to the same model family. Simply adopt a convolutional projector is able to align their features for distillation. However, heterogeneous models learn diverse features because of there distinct inductive biases. Under this circumstance, directly forcing features of the student to mimic the teacher may impede the performance. Moreover, ViTs use embedded image patches as input, and some architectures even adopt an additional classification token [11, 17], which further aggregates the feature mismatch problem.

**Centered kernel alignment analysis.**[2] To demonstrate the representation gap among heterogeneous architectures, we adopt centered kernel alignment (CKA) [48, 49] to compare features extracted by CNN, ViT, and MLP models. CKA is a feature similarity measurement allowing cross-architecture comparison achievable, as it can work with inputs having different dimensions.

CKA evaluates feature similarity over mini-batch. Suppose $\mathbf{X} \in \mathbb{R}^{n \times d_1}$ and $\mathbf{Y} \in \mathbb{R}^{n \times d_2}$ are features of $n$ samples extracted by two different models, where $d_1$ and $d_2$ are their dimensions, respectively. CKA measures their similarity by:

$$\text{CKA}(\mathbf{K}, \mathbf{L}) = \frac{\mathcal{D}_{\text{HSIC}}(\mathbf{K}, \mathbf{L})}{\sqrt{\mathcal{D}_{\text{HSIC}}(\mathbf{K}, \mathbf{K})\mathcal{D}_{\text{HSIC}}(\mathbf{L}, \mathbf{L})}}, \tag{3}$$

where $\mathbf{L} = \mathbf{X}\mathbf{X}^T$ and $\mathbf{K} = \mathbf{Y}\mathbf{Y}^T$ are Gram matrices of the features, and $\mathcal{D}_{\text{HSIC}}$ is the Hilbert-Schmidt independence criterion [50], a non-parametric independence measure. The empirical estimator of $\mathcal{D}_{\text{HSIC}}$ can be formulated as:

$$\mathcal{D}_{\text{HSIC}}(\mathbf{K}, \mathbf{L}) = \frac{1}{(n-1)^2}\text{tr}(\mathbf{K}\mathbf{H}\mathbf{L}\mathbf{H}), \tag{4}$$

where $\mathbf{H}$ is the centering matrix $\mathbf{H}_n = \mathbf{I}_n - \frac{1}{n}\mathbf{1}\mathbf{1}^T$.

Figure 1 shows feature similarity heatmaps measured by CKA. From figures at the first row, models of homogeneous architectures tend to learn similar features at layers of similar positions. On the country, heatmaps at the second row demonstrate that features of heterogeneous architectures are remarkably different. In particular, MobileNet v2 can only learn features similar to shallower layers of ViT-S or Mixer-B/16. Moreover, features of ViT-S and Mixer-B/16 are quite different as well.

From the CKA analysis, directly using hint-based distillation methods for cross-architecture KD is unreasonable. Although only using logits for distillation is feasible, the absence of supervision at intermediate layers may lead to suboptimal results. New designs of distilling at intermediate layers are needed to improve heterogeneous KD.

### 3.2 Generic heterogeneous knowledge distillation

To bridge the gap of heterogeneous architectures in KD, we propose a one-for-all cross-architecture KD method, which can perform KD between models of any architectures, such as CNN, ViT, and MLP. Figure 2 provides an overview of the proposed framework.

**Learning in the logits space.** Following the CKA analysis results, heterogeneous models learn distinct representations, which impedes the feature distillation process. Hence, the key for heterogeneous feature distillation is to align the mismatched representations. However, existing approaches

---

[2]Detailed settings of our CKA analysis are provided in our supplement.

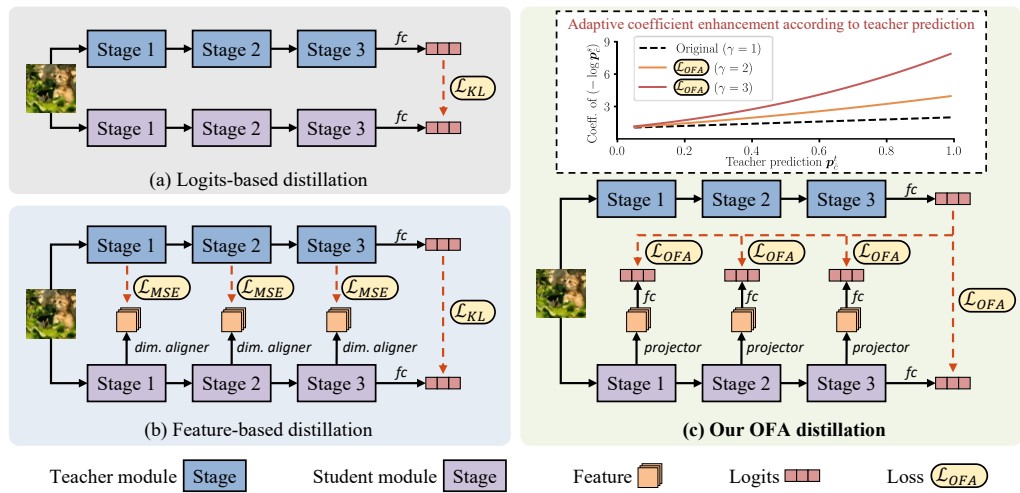

Figure 2: Comparison of different KD approaches. (a) Logits-based distillation: student only learns from final predictions of the teacher; (b) hint-based distillation: besides final predictions, student also learns to mimic intermediate features of the teacher; and (c) our OFA-KD framework: student learns from final predictions of the teacher via multiple branches, where the OFA distillation loss adaptively adjusts the amount of target information to be learned based on the confidence of teacher. *Note*: Only three stages are shown for convenience. In our experiment, all models are split into four stages.

directly align dimension of features via a convolutional projector [6], which is not a generic solution for all heterogeneous architectures. Inspired by early-exit model architecture [51], we align features in the logits space by introducing additional exit branches into the student. In particular, we build each exit branch with a feature projector and a classifier layer. On the teacher side, we direct adopt its final output as the knowledge. In the logits space, redundant architecture-specific information is removed, and thus feature alignment across any architectures is achievable. When training the student, the additional exit branches are optimized together with the student backbone. At test time, these branches are removed and render no extra inference overhead.

**Adaptive target information enhancement.** Although different models learn the same targets in the logits space, their different inductive bias leads them to variant destinations, *i.e.*, their predictive distributions are not the same. For example, a CNN model may predict a sample with similar probabilities of belonging to cars and trucks, as CNNs tend to catch the shared local information of the two classes, *e.g.*, round wheels. In contrast, ViTs may differentiate these samples better, as its attention mechanism is apt to learn the distinct global features, *e.g.*, arc-shaped and rectangle-shaped bodies. This difference results in an inconsistent understanding of the dark knowledge of the teacher and the student. To bridge this gap, we propose to enhance the information of the target class as it always relates to the real class of a sample and brings no concern about information mismatch.

We take the first step by reformulating the original distillation loss to decouple information about the target class:[3]

$$
\begin{aligned}
\mathcal{L}_{\mathrm{KD}} &= -\log \boldsymbol{p}_{\hat{c}}^s - \mathbb{E}_{c\sim\mathcal{Y}}[\boldsymbol{p}_c^t \log \boldsymbol{p}_c^s] \\
&= -(1+\boldsymbol{p}_{\hat{c}}^t)\log \boldsymbol{p}_{\hat{c}}^s - \mathbb{E}_{c\sim\mathcal{Y}/\{\hat{c}\}}[\boldsymbol{p}_c^t \log \boldsymbol{p}_c^s],
\end{aligned}
\tag{5}
$$

where we ignored the denominator term in KL-divergence as it does not contribute to the gradient. $c$ is a predicted class and $\hat{c}$ indicates the target class. To enhance information from the target class, we add a modulating parameter $\gamma \geq 1$ to the $(1+\boldsymbol{p}_{\hat{c}}^t)$ term:

$$
\mathcal{L}_{\mathrm{OFA}} = -(1+\boldsymbol{p}_{\hat{c}}^t)^\gamma \log \boldsymbol{p}_{\hat{c}}^s - \mathbb{E}_{c\sim\mathcal{Y}/\{\hat{c}\}}[\boldsymbol{p}_c^t \log \boldsymbol{p}_c^s].
\tag{6}
$$

---

[3]Without loss of generality, we omit the hyperparameter $\lambda$.

**Discussion.** To learn the behavior of the OFA loss with different $\gamma$ values, we take the case of $\gamma \in \mathbb{Z}_+$ as an example expand the binomial term:[4]

$$
\begin{aligned}
\mathcal{L}_{\text{OFA}} &= -\sum_{k=0}^{\gamma}\binom{\gamma}{k}(\boldsymbol{p}_{\hat{c}}^t)^k \log \boldsymbol{p}_{\hat{c}}^s - \mathbb{E}_{c\sim\mathcal{Y}/\{\hat{c}\}}[\boldsymbol{p}_c^t \log \boldsymbol{p}_c^s] \\
&= -(1+\boldsymbol{p}_{\hat{c}}^t)\log \boldsymbol{p}_{\hat{c}}^s - \mathbb{E}_{c\sim\mathcal{Y}/\{\hat{c}\}}[\boldsymbol{p}_i^t \log \boldsymbol{p}_i^s] - (\sum_{k=1}^{\gamma}\binom{\gamma}{k}(\boldsymbol{p}_{\hat{c}}^t)^k - \boldsymbol{p}_{\hat{c}}^t)\log \boldsymbol{p}_{\hat{c}}^s \qquad (7) \\
&= \mathcal{L}_{KD} + -(\sum_{k=1}^{\gamma}\binom{\gamma}{k}(\boldsymbol{p}_{\hat{c}}^t)^k - \boldsymbol{p}_{\hat{c}}^t)\log \boldsymbol{p}_{\hat{c}}^s.
\end{aligned}
$$

When $\gamma = 1$, the OFA distillation loss is identical to the logits-based KD loss. To study cases of $\gamma \geq 1$, we take $\gamma = 2$ as an example:

$$
\mathcal{L}_{\text{OFA},\gamma=2} = \mathcal{L}_{KD} + -(\boldsymbol{p}_{\hat{c}}^t + (\boldsymbol{p}_{\hat{c}}^t)^2)\log \boldsymbol{p}_{\hat{c}}^s. \qquad (8)
$$

Besides the KD loss term, the OFA distillation loss has an additional positive term that only relates to the target class. If the teacher is confident at the target class, the high order term $(\boldsymbol{p}_{\hat{c}}^t)^2$ decays slowly. Otherwise, the high order term decays faster, which prevents the student to learn from a less confident teacher and achieves adaptive enhancement of target information. The top-right corner of Figure 2 provides a brief illustration about effect of the high order term in the OFA loss.

Heterogeneous models inherently possess differences in their learning capabilities and preferences. Our proposed adaptive target information enhancement method bridges this gap by mitigating the impact of soft labels when the teacher provides suboptimal predictions. Moreover, the efficacy of distillation between heterogeneous architectures can be further enhanced by combining the multi-branch learning paradigm in the aligned logits space.

## 4 Experiment

### 4.1 Experimental setup

We conducted comprehensive experiments to evaluate the proposed OFA-KD framework. In this section, we offer a concise overview of our experimental settings. Additional details can be found in our supplementary materials.

**Models.** We evaluated a variety of models with heterogeneous architectures. Specifically, we employed ResNet [10], MobileNet v2 [18], and ConvNeXt [12] as CNN models, ViT [11], DeiT [17], and Swin [21] as ViT models, and MLP-Mixer [13] and ResMLP [24] as MLP models.

In our OFA-KD framework, incorporating additional exit branches into the student model requires identifying appropriate branch insertion points. For models with a pyramid structure, we consider the end of each stage as potential insertion points, resulting in a total of four points. In the case of other models like ViT, we evenly divide them into four parts and designate the end of each part as feasible insertion points for the branches. Exit branches in CNNs are constructed using depth-width convolutional layers, while exit branches in ViTs and MLPs employ ViT blocks.

When used as the student model, the smallest variant of Swin-T demonstrates superior performance compared to several teachers when trained from scratch. For example, Swin-T outperforms Mixer-B/16 on the ImageNet-1K dataset. To facilitate comparisons between such teacher-student combinations, we introduce two modified versions of Swin-T: Swin-Nano and Swin-Pico, referred to as Swin-N and Swin-P respectively.

**Datasets.** We adopt the CIFAR-100 dataset [52] and the ImageNet-1K dataset [53] for evaluation. CIFAR-100 comprises 50K training samples and 10K testing samples of $32\times32$ resolution, while the ImageNet-1K dataset is more extensive, containing 1.2 million training samples and 50,000 validation samples, all with a resolution of $224\times224$. Since ViTs and MLPs accept image patches as input, we upsample the images in CIFAR-100 to the resolution of 224x224 for all subsequent experiments to facilitate the patch embedding process.

**Baselines.** We employ both logits-based and hint-based knowledge distillation (KD) approaches as our baselines. Specifically, the hint-based approaches include FitNet [6], CC [54], RKD [37], and CRD [40], while the logits-based baselines consist of KD [5], DKD [55], and DIST [9].

---

[4]We take the assumption of $\gamma \in \mathbb{Z}_+$ above just for convenience, and there is no limitation of requiring $\gamma$ to be an integer in practice.

Table 1: KD methods with heterogeneous architectures on ImageNet-1K. The best results are indicated in bold, while the second best results are underlined. †: results achieved by combining with FitNet.

| Teacher | Student | From Scratch | | hint-based | | | | Logits-based | | | |
|---------|---------|------|------|--------|-------|-------|-------|-------|-------|-------|-------|
| | | T. | S. | FitNet | CC | RKD | CRD | KD | DKD | DIST | OFA |
| *CNN-based students* | | | | | | | | | | | |
| DeiT-T | ResNet18 | 72.17 | 69.75 | 70.44 | 69.77 | 69.47 | 69.25 | 70.22 | 69.39 | 70.64 | **71.34** |
| Swin-T | ResNet18 | 81.38 | 69.75 | 71.18 | 70.07 | 68.89 | 69.09 | 71.14 | 71.10 | 70.91 | **71.85** |
| Mixer-B/16 | ResNet18 | 76.62 | 69.75 | 70.78 | 70.05 | 69.46 | 68.40 | 70.89 | 69.89 | 70.66 | **71.38** |
| DeiT-T | MobileNetV2 | 72.17 | 68.87 | 70.95 | 70.69 | 69.72 | 69.60 | 70.87 | 70.14 | 71.08 | **71.39** |
| Swin-T | MobileNetV2 | 81.38 | 68.87 | 71.75 | 70.69 | 67.52 | 69.58 | 72.05 | 71.71 | 71.76 | **72.32** |
| Mixer-B/16 | MobileNetV2 | 76.62 | 68.87 | 71.59 | 70.79 | 69.86 | 68.89 | 71.92 | 70.93 | 71.74 | **72.12** |
| *ViT-based students* | | | | | | | | | | | |
| ResNet50 | DeiT-T | 80.38 | 72.17 | 75.84 | 72.56 | 72.06 | 68.53 | 75.10 | 75.60† | 75.13† | **76.55†** |
| ConvNeXt-T | DeiT-T | 82.05 | 72.17 | 70.45 | 73.12 | 71.47 | 69.18 | 74.00 | 73.95 | 74.07 | **74.41** |
| Mixer-B/16 | DeiT-T | 76.62 | 72.17 | 74.38 | 72.82 | 72.24 | 68.23 | 74.16 | 72.82 | 74.22 | **74.46** |
| ResNet50 | Swin-N | 80.38 | 75.53 | 78.33 | 76.05 | 75.90 | 73.90 | 77.58 | 78.23† | 77.95† | **78.64†** |
| ConvNeXt-T | Swin-N | 82.05 | 75.53 | 74.81 | 75.79 | 75.48 | 74.15 | 77.15 | 77.00 | 77.25 | **77.50** |
| Mixer-B/16 | Swin-N | 76.62 | 75.53 | 76.17 | 75.81 | 75.52 | 73.38 | 76.26 | 75.03 | 76.54 | **76.63** |
| *MLP-based students* | | | | | | | | | | | |
| ResNet50 | ResMLP-S12 | 80.38 | 76.65 | 78.13 | 76.21 | 75.45 | 73.23 | 77.41 | 78.23† | 77.71† | **78.53†** |
| ConvNeXt-T | ResMLP-S12 | 82.05 | 76.65 | 74.69 | 75.79 | 75.28 | 73.57 | 76.84 | 77.23 | 77.24 | **77.53** |
| Swin-T | ResMLP-S12 | 81.38 | 76.65 | 76.48 | 76.15 | 75.10 | 73.40 | 76.67 | 76.99 | 77.25 | **77.31** |

**Optimization.** In our implementation, we employ different optimizers for training the student models based on their architecture. Specifically, all CNN students are trained using the SGD optimizer, while those with a ViT or MLP architecture are trained using the AdamW optimizer. For the CIFAR-100 dataset, all models are trained for 300 epochs. When working with the ImageNet-1K dataset, CNNs are trained for 100 epochs, whereas ViTs and MLPs are trained for 300 epochs.

## 4.2 Distillation results on ImageNet-1K

We first conduct experiments on the ImageNet-1K dataset. To make a comprehensive comparison, we adopt five student models and five teacher models belonging to three different model architecture families, and evaluate KD approaches on fifteen heterogeneous combinations of teacher and student. We report the results in Table 1.

In comparison to previous KD methods, our distillation framework achieves remarkable performance improvements. Specifically, when training a student based on CNN architecture, our proposed method achieves performance improvements ranging from 0.20% to 0.77% over the second-best baselines. When the student model is based on a ViT or MLP architecture, which utilizes image patches as input, our OFA-KD also achieves remarkable performance improvement, with a maximum accuracy gain of 0.71%. Significantly, the hint-based baseline FitNet exhibits noteworthy performance when the teacher model is ResNet50 and the student model is a ViT or MLP. We speculate that designs based on traditional 3×3 convolution of ResNet50 have the capability to capture local information, such as texture details, better than ViT/MLP. In the process of "patchify", ViT/MLP might overlook these local details. Therefore, when ResNet50 serves as the teacher, FitNet can provide valuable intermediate representations. We thus report the results of logit-based methods obtained by combining FitNet and OFA-KD when adopting ResNet50 as the teacher.

Based on our experimental results, we observe that the second-best performance is not consistently achieved by the same baseline approach. FitNet, vanilla KD, and DIST take turns in attaining the second-best results, indicating that previous KD approaches face challenges in consistently improving performance when distilling knowledge between heterogeneous architectures. However, our proposed approach consistently outperforms all baselines in all cases, showcasing its effectiveness in cross-architecture distillation.

Table 2: KD methods with heterogeneous architectures on CIFAR-100. The best results are indicated in bold, while the second best results are underlined.

| Teacher | Student | From Scratch | | hint-based | | | | Logits-based | | | |
|---|---|---|---|---|---|---|---|---|---|---|---|
| | | T. | S. | FitNet | CC | RKD | CRD | KD | DKD | DIST | OFA |
| *CNN-based students* | | | | | | | | | | | |
| Swin-T | ResNet18 | 89.26 | 74.01 | 78.87 | 74.19 | 74.11 | 77.63 | 78.74 | 80.26 | 77.75 | **80.54** |
| ViT-S | ResNet18 | 92.04 | 74.01 | 77.71 | 74.26 | 73.72 | 76.60 | 77.26 | 78.10 | 76.49 | **80.15** |
| Mixer-B/16 | ResNet18 | 87.29 | 74.01 | 77.15 | 74.26 | 73.75 | 76.42 | 77.79 | 78.67 | 76.36 | **79.39** |
| Swin-T | MobileNetV2 | 89.26 | 73.68 | 74.28 | 71.19 | 69.00 | 79.80 | 74.68 | 71.07 | 72.89 | **80.98** |
| ViT-S | MobileNetV2 | 92.04 | 73.68 | 73.54 | 70.67 | 68.46 | 78.14 | 72.77 | 69.80 | 72.54 | **78.45** |
| Mixer-B/16 | MobileNetV2 | 87.29 | 73.68 | 73.78 | 70.73 | 68.95 | 78.15 | 73.33 | 70.20 | 73.26 | **78.78** |
| *ViT-based students* | | | | | | | | | | | |
| ConvNeXt-T | DeiT-T | 88.41 | 68.00 | 60.78 | 68.01 | 69.79 | 65.94 | 72.99 | 74.60 | 73.55 | **75.76** |
| Mixer-B/16 | DeiT-T | 87.29 | 68.00 | 71.05 | 68.13 | 69.89 | 65.35 | 71.36 | 73.44 | 71.67 | **73.90** |
| ConvNeXt-T | Swin-P | 88.41 | 72.63 | 24.06 | 72.63 | 71.73 | 67.09 | 76.44 | 76.80 | 76.41 | **78.32** |
| Mixer-B/16 | Swin-P | 87.29 | 72.63 | 75.20 | 73.32 | 70.82 | 67.03 | 75.93 | 76.39 | 75.85 | **78.93** |
| *MLP-based students* | | | | | | | | | | | |
| ConvNeXt-T | ResMLP-S12 | 88.41 | 66.56 | 45.47 | 67.70 | 65.82 | 63.35 | 72.25 | 73.22 | 71.93 | **81.22** |
| Swin-T | ResMLP-S12 | 89.26 | 66.56 | 63.12 | 68.37 | 64.66 | 61.72 | 71.89 | 72.82 | 11.05 | **80.63** |

## 4.3 Distillation results on CIFAR-100

In addition to the ImageNet-1K dataset, we also evaluate the proposed method on the CIFAR-100 dataset. We conduct experiments with twelve combinations of heterogeneous teacher and student models, and report the results in Table 2.

On this small-scale dataset, the hint-based approaches exhibit inferior performance, particularly when the student model is of ViT or MLP architecture, highlighting its limitations in distilling knowledge across heterogeneous architectures. For instance, FitNet only achieves a 24.06% accuracy when used in combination with a ConvNeXt-T teacher and a Swin-P student. By contrast, the OFA-KD method continues to achieve significant performance improvements, ranging from 0.28% to 8.00% over the second-best baselines.

Notably, when evaluating on the ImageNet-1K dataset, DIST achieves the second-best results at most times. while on the CIFAR-100 dataset, DKD tends to achieve the second-best results more frequently. This is because DIST relaxes the strict prediction imitation by correlation mimicry, which is more advantageous for learning from a stronger teacher trained on ImageNet-1K. On the other hand, DKD enhances the dark knowledge in teacher predictions, which proves beneficial when working with a smaller teacher trained on CIFAR-100. However, our OFA-KD method can enhance target information adaptively, enabling the student model to consistently achieve the best performance.

## 4.4 Ablation study

**Number and position of exit branches.** In our OFA-KD method, branches are integrated into the student to facilitate learning at intermediate layers. We conduct experiments to learn the impact of the number and position of branches. As the student models are divided into four stages, we use numbers 1 to 4 to indicate the insertion points of branches. As shown in Table 3, when comparing learning at the last layer with learning from intermediate layers, the ResNet18 student presents no preference to the position of the branch, as they all help improve the performance. However, DeiT-T only performs better when branches are added at the end of stage 1 or stage 4. In the evaluation of using multiple branches. The results demonstrate that learning at the end of all the four stages is the best configuration for the OFA-KD method.

Table 3: Impact of exit branch number and position on ImageNet-1K. ∅: only learning at the last layer.

| Stage | T: DeiT-T S: ResNet18 | T: ResNet50 S: DeiT-T |
|---|---|---|
| ∅ | 70.62 | 75.18 |
| {1} | 70.75 | 75.35 |
| {2} | 70.70 | 75.21 |
| {3} | 70.78 | 75.17 |
| {4} | 70.78 | 75.28 |
| {1,4} | 71.04 | 75.52 |
| {1,2,4} | 71.08 | 75.61 |
| {1,2,3,4} | **71.34** | **75.73** |

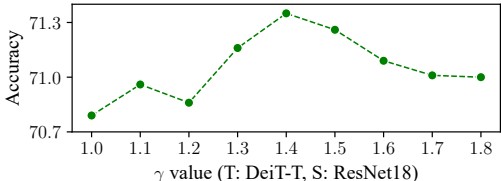 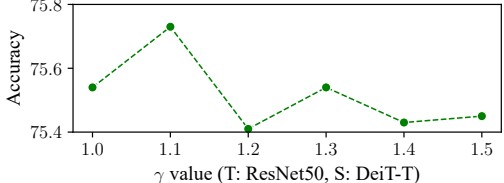

Figure 3: Ablation study on different modulating parameter $\gamma$ settings on ImageNet-1K.

Table 5: KD methods with homogeneous architectures on ImageNet-1K. *T*: ResNet34, *S*: ResNet18.

|  | T. | S. | KD [5] | OFD [56] | Review [43] | CRD [37] | DKD [55] | DIST [9] | OFA |
|---|---|---|---|---|---|---|---|---|---|
| Accuracy | 73.31 | 69.75 | 70.66 | 70.81 | 71.61 | 71.17 | 71.70 | 72.07 | **72.10** |

Table 6: Comparison of homogeneous and heterogeneous teacher on ImageNet-1K.

| Teacher | T. | S.(ResNet50) | RKD [37] | Review [43] | CRD [37] | DKD [55] | DIST [9] | OFA |
|---|---|---|---|---|---|---|---|---|
| ResNet152 | 82.83 | 79.86 | 79.53 | 80.06 | 79.33 | 80.49 | 80.55 | 80.64 |
| ViT-B | 86.53 | 79.86 | 79.38 | 79.32 | 79.48 | 80.76 | 80.90 | 81.33 |

**Scale of OFA loss.** The modulation parameter is introduced at the exponential position of the term $1 + \boldsymbol{p}_{\hat{c}}^{t}$, which has a value greater than 1. This increases the overall loss scale. To ensure a stable training process, we compare two schemes: multiplying a scaling factor to the OFA loss and clipping gradient by setting a maximum norm value. As shown in Table 4, the best configuration is setting the scale factor to 1 and the maximum gradient norm to 5. Considering that searching the optimal scale factor and gradient clipping settings for each combination of teacher and student is expensive, we adopt the aforementioned best settings in all experiments.

Table 4: Comparison of scaling factors and clip grad settings on ImageNet-1K. The scaling factors are directly multiplied to the OFA loss. We adopt a ResNet50 teacher and a DeiT-T student with $\gamma = 1.1$.

| Scaling factor | Acc. | Clip grad | Acc. |
|---|---|---|---|
| 0.5 | 75.29 | 1 | 75.39 |
| 0.8 | 75.58 | 3 | 75.61 |
| 1.0 | **75.73** | 4 | 75.52 |
| 1.2 | 75.54 | 5 | **75.73** |
| 1.4 | 75.64 | 6 | 75.32 |
| 1.6 | 75.46 | 7 | 75.36 |

**Modulating parameter.** The modulation parameter $\gamma$ controls the strength of the adaptive target information enhancement. To investigate its impact, we compare different configurations of the $\gamma$ value setting on ImageNet-1K and present the results in Figure 3. Specifically, when the teacher model is DeiT-T and the student model is ResNet18, the best student performance is achieved when $\gamma$ is set to 1.4. For the teacher-student pair of ResNet50 and DeiT-T, the optimal configuration for $\gamma$ is 1.1. This difference in optimal $\gamma$ values can be attributed to the varying strengths of the teacher models. As DeiT-T is a weaker teacher model compared to ResNet50, a larger $\gamma$ value is required to mitigate the additional interference information. By increasing the $\gamma$ value, the adaptive target information enhancement compensates for the limitations of the weaker teacher model, enabling the student model to better utilize the available information and improve its performance.

**Distillation in homogeneous architectures.** We adopt the commonly used combination of ResNet34 teacher and ResNet18 student on ImageNet-1K to further demonstrate the effectiveness of our method across homogeneous architectures. As the results shown in Table 5, the OFA method achieves performance on par with the best distillation baseline.

**Homogeneous *vs.* heterogeneous teachers** To assess the impact of utilizing a larger heterogeneous teacher model, we train a ResNet50 student with both a ResNet152 teacher (homogeneous architecture) and a ViT-B teacher (heterogeneous architecture). As indicated in Table 6, our OFA-KD approach achieves a notable 0.41% performance improvement when utilizing the ViT-B teacher compared to the ResNet152 teacher. This result underscores the significance of cross-architecture knowledge distillation (KD) in pursuing better performance improvement.

# 5   Conclusion

This paper studies how to distill heterogeneous architectures using hint-based knowledge. We demonstrate that the distinct representations learned by different architectures impede existing feature distillation approaches, and propose a one-for-all framework to bridge this gap in cross-architecture KD. Specifically, we introduce exit branches into the student to project features into the logits space and train the branches with the logits output of the teacher. The projection procedure discards architecture-specified information in feature, and thus hint-based distillation becomes feasible. Moreover, we propose a new loss function for cross-architecture distillation to adaptive target information enhancement, which is achieved by adding a modulating parameter to the reformulated KD loss. We conduct extensive experiments on CIFAR-100 and ImageNet-1K. Corresponding results demonstrate the effectiveness of the proposed cross-architecture KD method.

**Limitation.** While our approach achieves remarkable performance improvements in distilling ResNet50, it is worth noting that for some architectures, *e.g.*, ResNet18, the performance distilled by a heterogeneous teacher is lower than that achieved by a homogeneous teacher. Furthermore, the requirement of finding an optimal setting for the modulating parameter adds a level of complexity. Improperly setting this parameter could potentially lead to inferior performance.

**Border impact.** Our work mainly studies hint-based distillation for heterogeneous architecture and proposes a generic framework. The performance improvement achieved by our OFA-KD demonstrates that there exists useful information between heterogeneous architectures. We hope that our research will bring new perspective to KD community for seeking better aligning approaches and inspire future works for cross-architecture distillation.

# Acknowledgement

This work is supported by National Key Research and Development Program of China under No. SQ2021YFC3300128 National Natural Science Foundation of China under Grant 61971457. And Chang Xu was supported in part by the Australian Research Council under Projects DP210101859 and FT230100549.

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
