# One-for-All: Bridge the Gap of Heterogeneous Architectures in Knowledge Distillation Supplementary Material

**Zhiwei Hao**[1,2]**, Jianyuan Guo**[3]**, Kai Han**[2]**, Yehui Tang**[2]**,**
**Han Hu**[1*]**, Yunhe Wang**[2*]**, Chang Xu**[3]

[1]School of information and Electronics, Beijing Institute of Technology.
[2]Huawei Noah's Ark Lab.
[3]School of Computer Science, Faculty of Engineering, The University of Sydney.

{haozhw, hhu}@bit.edu.cn, jguo5172@uni.sydney.edu.au,
{kai.han, yehui.tang, yunhe.wang}@huawei.com, c.xu@sydney.edu.au

## A    Necessity of heterogeneous teachers

To further emphasize the importance of leveraging heterogeneous teachers, we evaluate the effectiveness of our OFA-KD framework using four MLP-based models: Mixer-B/16 [1], ViP-S/7 [2], CycleMLP-B3 [3], and HireMLP-S [4]. As the current best MLP-based model achieves only 83.8% top-1 accuracy on ImageNet-1K validation set, it is challenging to find a superior MLP-based teacher model for further performance enhancement. Consequently, we employ a ViT-B model with a top-1 accuracy of 86.53% as the teacher model for conducting cross-architecture distillation. The comparison between models trained using the OFA-KD framework and models trained from scratch is illustrated in Figure 5 (the results of ResNet50 are also included in our main paper).

Our OFA-KD framework exhibits significant enhancements in terms of top-1 accuracy when compared to models trained from scratch. The observed improvements range from 1.2% to 2.6%. Notably, even the mid-size CycleMLP-B3 model (considering that the largest CycleMLP model is CycleMLP-B5) achieves an impressive accuracy of 84.08%, surpassing the accuracy achieved by larger and more advanced MLP models [1, 2, 3, 4, 5, 6]. This highlights the effectiveness of our framework in leveraging cross-architecture distillation to achieve superior performance.

## B    Details of experimental setup

### B.1    CKA analysis

In the Centered Kernel Alignment (CKA) analysis, we evaluate the learned features of three pretrained models: MobileNet v2 [7], ViT-S [8], and Mixer-B/16 [1], all of which were trained on the ImageNet-1K dataset. To conduct the analysis, we select a batch of 128 samples from the ImageNet-1K validation set and collect model activations after the activation layers. In order to simplify the matrix manipulation computations, we average each feature over its spatial dimensions before calculating the CKA similarity. More results of CKA analysis are provided in Figure 6.

### B.2    Architecture of Swin-Nano and Swin-Pico

To ensure that the teacher models outperform the student model, we introduce two modified versions of Swin-Tiny [9] called Swin-Nano and Swin-Pico. Swin-Nano has an embedding dimension of 64, while Swin-Pico has an embedding dimension of 48, compared to the original Swin-Tiny's embedding

---

*Corresponding author.

37th Conference on Neural Information Processing Systems (NeurIPS 2023).

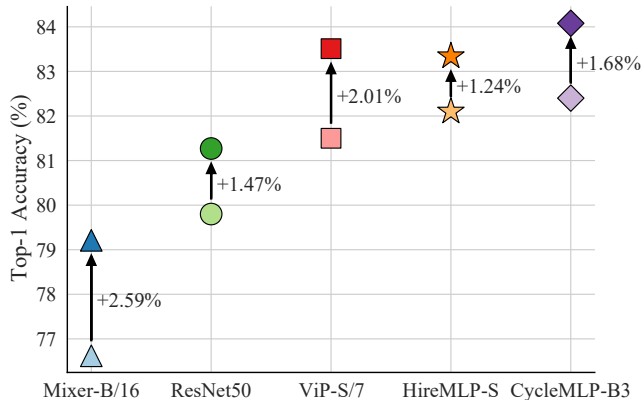

Figure 5: Comparison of model trained from scratch and models trained using our OFA-KD on ImageNet-1K. The teacher model is ViT-B with a top-1 accuracy of 86.53%. Light-colored markers represent results obtained by training from scratch, while dark-colored markers represent results obtained using OFA-KD. Black arrows indicate performance improvements, with the corresponding numerical results displayed alongside.

Table 8: Details of optimization settings.

|  | ImageNet-1K | | CIIFAR-100 | |
| --- | --- | --- | --- | --- |
|  | CNN | ViT/MLP | CNN | ViT/MLP |
| Epochs | 100 | 300 | 300 | 300 |
| Batch size | 512 | 1024 | 1024 | 1024 |
| Initial LR | 0.1 | 5e-4 | 5e-2 | 5e-4 |
| Minimum LR | 1e-6 | 1e-6 | 1e-3 | 1e-5 |
| Optimizer | SGD | AdamW | SGD | AdamW |
| Weight decay | 1e-4 | 5e-2 | 2e-3 | 5e-2 |
| LR schedule | ×0.1 at [30,60,90] | Cosine | Cosine | Cosine |
| Warmup | 3 | 20 | 3 | 20 |
| EMA [10] | - | 0.99996 | - | - |
| RandAugment [11] | - | 9/0.5 | - | 9/0.5 |
| Mixup [12] | - | 0.8 | - | 0.8 |
| Cutmix [13] | - | 1.0 | - | 1.0 |
| RE prob [14] | - | 0.25 | - | 0.25 |

dimension of 96. Moreover, Swin-Tiny has depths of (2, 2, 6, 2) and num_heads of (3, 6, 12, 24), while the two modified models have the same configuration for depths and num_heads, which are (2, 2, 2, 2) and (2, 4, 8, 16), respectively.

## B.3 Optimization

For training models of various architectures on the ImageNet-1K and CIFAR-100 datasets, we use different optimization settings. The detailed settings can be found in Table 8.

## B.4 Modification of baselines

Since FitNet is designed for CNN models, we slightly adapt its feature alignment process for cross-architecture distillation. When comparing features between a CNN model and a ViT or MLP model, we first project the features of both models into the embedding space. This ensures that all features are represented in the same embedding space. Next, we align the patch number of the features using the patch merging block from Swin [9]. Finally, we align the embedding dimension using a linear layer. For comparison between ViT and MLP models, only the above last two steps are required.

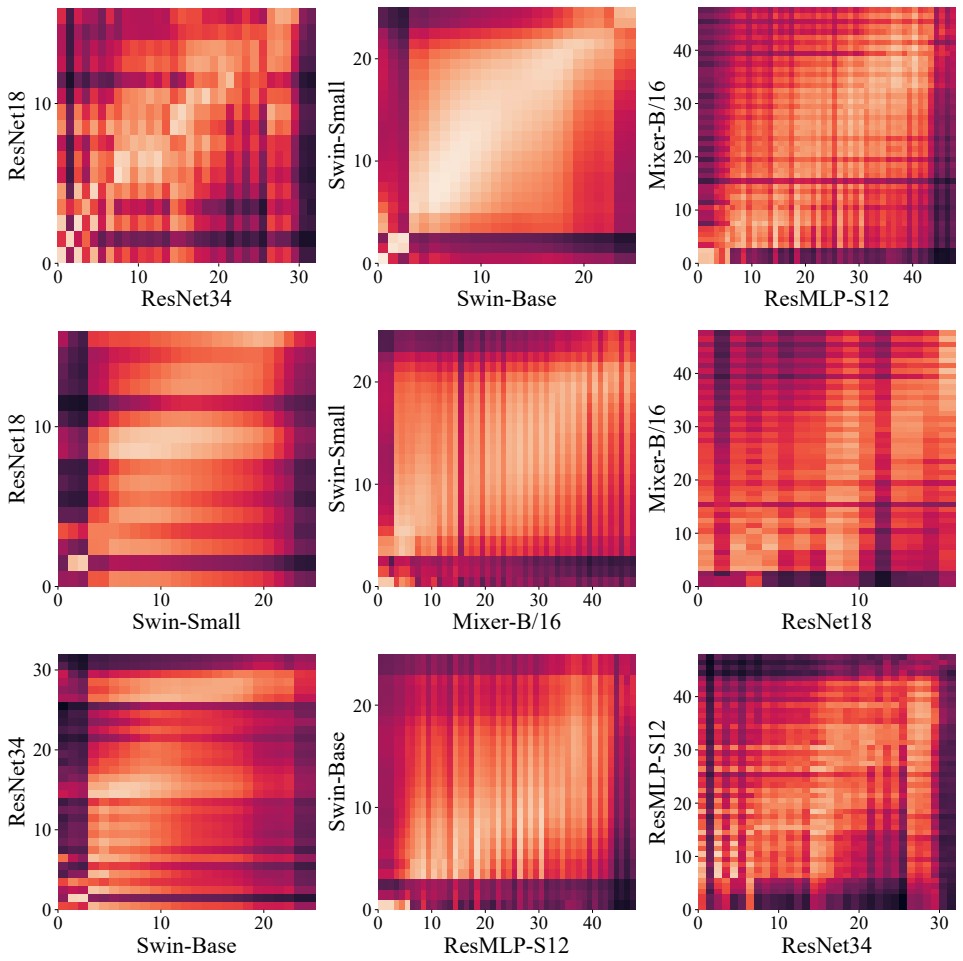

Figure 6: Additional CKA analysis results. Heatmaps in the first row compares intermediate features of homogeneous models while heatmaps in the last two rows compare intermediate features of heterogeneous models.

## B.5 Branch architecture

In our OFA-KD framework, we introduce additional branches to enable training the student at intermediate layers. These branches are composed of depth-width convolutional blocks for CNN students and vision transformer blocks for ViT or MLP students. Table 9 provides PyTorch-style pseudocode for constructing these branches using timm framework [15].

Table 9: PyTorch-style pseudocode for building branches.

```python
## CNN students, original feature shape = (in_chans, H, W)
if stage != 4:
    down_sample_blk_num = 4 - stage
    down_sample_blks = []
    for i in range(down_sample_blk_num):
        if i == down_sample_blk_num - 1:
            # dimension of features at penultimate layer
            out_chans = max(feature_dim_s, feature_dim_t)
        else:
            out_chans = in_chans * 2
        #
        down_sample_blks.append(
            timm.models.layers.SeparableConvNormAct(in_chans, out_chans))
        in_chans *= 2
else:
    down_sample_blks = [nn.Conv2d(
        in_chans, max(feature_dim_s, feature_dim_t), 1, 1, 0)]

branch = nn.Sequential(
    *down_sample_blks,
    nn.AdaptiveAvgPool2d(1),
    nn.Flatten(),
    nn.Linear(max(feature_dim_s, feature_dim_t), num_classes))

## ViT/MLP students, original feature shape = (patch_num, embed_dim)
final_patch_grid = 7  # there are 49 patches after merging
patch_grid = int(patch_num ** .5)
merge_num = max(int(np.log2(patch_grid / final_patch_grid)), 0)
merger_modules = []
for i in range(merge_num):
    if i == 0:
        merger_modules.append(
            timm.models.layers.PatchMerging(
                input_resolution=(patch_grid // 2 ** i, patch_grid // 2 ** i),
                dim=embed_dim,
                out_dim=feature_dim_s,
                act_layer=nn.GELU))
    else:
        merger_modules.append(
            timm.models.layers.PatchMerging(
                input_resolution=(patch_grid // 2 ** i, patch_grid // 2 ** i),
                dim=feature_dim_s,
                out_dim=feature_dim_s,
                act_layer=nn.GELU if i != merge_num - 1 else nn.Identity))
patch_merger = nn.Sequential(*merger_modules)
blocks = nn.Sequential(
    *[timm.models.layers.Block(
        dim=feature_dim_s, num_heads=4) for _ in range(max(4 - stage, 1))]
)
branch = nn.Sequential(
    patch_merger,
    blocks,
    global_pool,  # x = x.mean(dim=1)
    nn.Linear(feature_dim_s, args.num_classes))
```