# OpenReview forum: "One-for-All: Bridge the Gap Between Heterogeneous Architectures in Knowledge Distillation"
_NeurIPS.cc/2023/Conference — NeurIPS 2023 poster_

### Official Review · Reviewer_FwZi · 2023-07-05

**Soundness:** 2 fair
**Presentation:** 2 fair
**Contribution:** 3 good
**Rating:** 5
**Confidence:** 5

**Summary:**

In this paper, the authors propose a new heterogeneous knowledge distillation approach. The core idea is to map the intermediate layer features of the network to a unified logit space to eliminate feature mismatches caused by different structures. The author conducts thorough experiments on distilling between CNN, ViT, and MLP networks. According to the experimental results in the paper, the proposed approach yields promising results.

**Strengths:**

1. The experiments in this paper are comprehensive, considering distillation between various networks with different structures, and conducting experiments on both CIFAR-100 and ImageNet datasets.
2. The proposed method is reasonable, as projecting network features onto a latent space to avoid the alignment issue of distillation between networks with different structures may indeed lead to better results.
3. According to the experimental results, the proposed method achieves good performance in various experiments.

**Weaknesses:**

1. The newly proposed method is very similar to deep supervision in that both involve adding an auxiliary head to the intermediate layer to learn the final output. The only difference is that deep supervision previously learned hard labels, while the proposed method learns soft labels from the teacher. However, this paper does not discuss the differences between this method and deep supervision, including theoretical and experimental results.
2. In some experimental settings, the improvement brought by the proposed method is very limited. And the comparison is not comprehensive. For example, in Table 1 and Table 2, OFD, Review, and CRD's results are missing.
3. In addition, in Table 1, some experiments are combined with FitNet while others are not. Although the authors have provided an explanation, it still feels strange that we cannot conclude that ResNet50's features are more applicable just because it is the most commonly used network. And why ResNet50 is not adopted as the teacher in Table 2?

**Questions:**

Please address the problem in Weakness

**Limitations:**

Potential negative societal is not applicable.

---

> ### Author Rebuttal · Authors · 2023-08-09
>
> > **Weakness 1:** The newly proposed method is very similar to deep supervision
>
> **Response to the weakness 1:** Thanks for your suggestions in improving quality of our work. Deep supervision [1,2,3] introduces intermediate supervision during training to mitigate the gradient vanishing/exploding problem in early era of deep learning. Conversely, feature distillation involves transferring knowledge at intermediate layers, aiming to equip the student with a more comprehensive understanding of the teacher's knowledge. Our OFA method falls within this category of approaches. To conduct a comparison between deep supervision method an OFA, we trained a ResNet18 model using deep supervision by replacing the OFA loss with a hard-label cross-entropy loss. We evaluated its performance against that achieved by OFA with a DeiT-T teacher. Notably, the position and number of auxiliary branches used in deep supervision remained consistent with OFA.
>
> |Teacher|Student|Method|Top-1|
> |:-:|:-:|:-:|:-:|
> |DeiT-T|ResNet18|OFA|71.34|
> |-|ResNet18|Deepsupervision|70.06|
>
> Based on the results above, it is evident that the model trained using our OFA framework outperforms the model trained with deep supervision by a significant margin. This disparity in performance underscores the superiority of our OFA in comparison to deep supervision.
>
> [1] Wang, Liwei, et al. "Training deeper convolutional networks with deep supervision."
>
> [2] Zhang, Linfeng, et al. "Contrastive deep supervision."
>
> [3] Li, Renjie, et al. "A comprehensive review on deep supervision: Theories and applications."
>
> > **Weakness 2:** In some experiment improvement is limited. Comparison is not comprehensive.
>
> **Response to the weakness 2:** Thanks for your valuable comments. Referring to the results in our main paper, our OFA method exhibits only a slight performance gain over the **second-best** baseline across certain teacher-student combinations. Notably, the second-best result isn't consistently obtained through the same baseline, signaling the challenges associated with generalizing existing methods to the context of heterogeneous KD. In contrast, our OFA method consistently outperforms all other baselines across all scenarios, yielding satisfactory results. This consistently strong performance underscores its generic applicability for cross-architecture KD.
>
> The OFD method is tailored for CNN models, leveraging features between Batch Normalization (BN) and ReLU layers for distillation. However, ViT/MLP architectures generally lack such intermediary positions, posing a challenge to the application of OFD. Furthermore, previous works like DIST and DKD have demonstrated their superiority over OFD. Hence, we have opted not to include OFD in our analysis.
>
> To assess the performance of Review, we selected two teacher-student pairs and conducted experiments on the ImageNet-1K dataset. The outcomes, as illustrated in the table below, reveal that Review falls short of our OFA methods by a noticeable margin. Notably, Review mandates features with dimensions of (N, C, H, W), where N, C, H, and W denote batch size, the number of channel, height, and width, respectively. Since ViT-generated features do not possess this structure, an "unpatchify" operation is required to transform them. We hypothesize that this transformation contributes to the less-than-ideal results obtained using Review. We intend to incorporate this discussion into our final revision.
>
> |Teacher|Student|KD|Review|OFA|
> |:-:|:-:|:-:|:-:|:-:|
> |DeiT-T|ResNet18|70.22|70.28|71.34|
> |ConvNeXt-T|DeiT-T|74.00|68.10|74.41|
>
> As for CRD, we have conducted the missed experiments on CIFAR-100, as shown in the following table. And we will include these results to our final revision.
>
> |Teacher|Student|CRD|
> |:-:|:-:|:-:|
> |Swin-T|ResNet18|77.63|
> |ViT-S|ResNet18|76.60|
> |Mixer-B/16|ResNet18|76.42|
> |Swin-T|MobileNetV2|79.80|
> |ViT-S|MobileNetV2|78.14|
> |Mixer-B/16|MobileNetV2|78.15|
> |ConvNeXt-T|DeiT-T|65.94|
> |Mixer-B/16|DeiT-T|65.35|
> |ConvNeXt-T|Swin-P|67.09|
> |Mixer-B/16|Swin-P|67.03|
> |ConvNeXt-T|ResMLP-S12|63.35|
> |Swin-T|ResMLP-S12|61.72|
>
> > **Weakness 3:** In Table 1, some experiments are combined with FitNet while others are not. Why ResNet50 is not adopted in Table 2?
>
> **Response to the weakness 3:** Thanks for your professional concern. To ensure a more fair comparison, we have chosen the two most competitive baselines, i.e., DKD and DKD, and have integrated them with FitNet to train the student. The outcomes are illustrated in the table below.
>
> |Teacher|Student|DKD+FitNet|DIST+FitNet|OFA+FitNet|
> |:-:|:-:|:-:|:-:|:-:|
> |ResNet50|DeiT-T|75.60|75.13|76.55|
> |ResNet50|Swin-N|78.23|77.95|78.64|
> |ResNet50|ResMLP-S12|78.23|77.71|78.53|
>
> The results indicate that DKD and DIST achieve satisfactory performance when integrated with FitNet. Nevertheless, our OFA method surpasses them, thereby highlighting the effectiveness of our approach.
>
> We speculate that designs based on traditional 3x3 convolution like ResNet50 have the capability to capture local information, such as texture details, better than ViT/MLP. In the process of "patchify", ViT might overlook these local details. Therefore, when ResNet50 serves as the teacher, FitNet can provide valuable intermediate representations. And we will correct the original statements in Line 244-246 of the main paper.
>
> Regarding the selection of ResNet50 as the teacher in Table 2 (experiments conducted on CIFAR-100), we conduct experiments to compare the performances between FitNet and OFA on two heterogeneous teacher-student pairs, i.e., ResNet-DeiT and ResNet-ResMLP.
>
> |Teacher|Student|FitNet|OFA|OFA+FitNet|
> |:-:|:-:|:-:|:-:|:-:|
> |ResNet50|DeiT-T|75.62|75.88|76.14|
> |ResNet50|ResMLP-S12|73.67|74.38|74.50|
>
> As the results shown in the above table, the integration of OFA and FitNet yields further enhancement in the performance of the student model. We will include the remaining omitted results, where ResNet50 serves as the teacher, and incorporate them into our final revision.

---

### Official Review · Reviewer_R7Hg · 2023-07-05

**Soundness:** 2 fair
**Presentation:** 2 fair
**Contribution:** 2 fair
**Rating:** 6
**Confidence:** 5

**Summary:**

This paper introduces a new method to distill knowledge between heterogeneous models named OFD-KD. This paper proposes to project the intermediate features into logits for distillation. A new loss function is also introduced in this paper to adaptively enhance the target information. Extensive experiments verify the effectiveness of this method.

**Strengths:**

1. This paper is easy to understand and clearly written.
2. This paper uses CKA to visualize the differences between CNN, VIT, and MLP.
3. Extensive experiments are conducted to verify the effectiveness of this method.

**Weaknesses:**

1. The improvement is relatively minor in ImageNet-1K. The most improvement is around 0.1%-0.3% in Tables 1, 5, and 6. This method does not exhibit a clear advantage over other techniques.
2. The architecture employed in this work has been extensively explored. This form of intermediate logit supervision has been widely used in BYOT, DCM, DKS, and other methods. This study does not offer any significant novelty in the context of distillation.
3. Table 1 fails to compare the latest feature-based distillation method, particularly SemCKD, which is a method dedicated to heterogeneous distillation. Thus, the experimental comparisons presented in this paper lack meaningfulness. The authors should compare with the recent state-of-the-art hint-based methods.
4. The author's use of CKA comparison is unnecessary, given the obvious architectural differences between CNN, MLP, and VIT. The differences between these architectures have been discussed in many works.
5. Please provide the results of OFA during distillation between heterogeneous networks such as VGG, ResNet, ShuffleNet, and MobileNet.

**Questions:**

1. In Tables 1 and 2, the results of many distillation methods are not as high as the baseline results, please explain your implementation details and why this is the case.
2. Is there any quantitative indicator to prove that the method proposed in this paper really bridges the gap of heterogeneous distillation?


**Limitations:**

See weaknesses and questions.

---

> ### Author Rebuttal · Authors · 2023-08-09
>
> # Response to Reviewer R7Hg part (1/2)
>
> **Response to the weakness 1:** In Table 5, the improvement when compared to methods tailored for homogeneous architectures, such as ResNet34-ResNet18, is indeed marginal. However, when evaluating heterogeneous pairs, as shown in Table 1, our OFA-KD consistently outperforms the second-best baseline by a range of 0.1% to 0.7%. Notably, it's worth noting that the second-best result is not consistently obtained by the same baseline.
>
> Given the intricate nature of cross-architecture KD, where architecture-specific nuances can hinder student learning, previous methods have struggled to consistently achieve satisfactory results. For instance, while DIST achieves the second-best performance across multiple teacher-student combinations, it slightly lags behind our proposed method. Nonetheless, on certain other teacher-student pairs, like the case of Swin-T teacher and ResNet18 student, DIST's performance significantly trails that of OFA. In contrast, OFA consistently attains the top performance, underscoring its applicability as a universal solution for both heterogeneous and homogeneous KD scenarios.
>
> **Response to the weakness 2:** Intermediate supervision is widely used in hint-based learning approaches like FitNet. BYOT and DCM's final structures, with the addition of auxiliary classifiers, bear similarities to our OFA. However, our focus is primarily on cross-architecture KD. Given the notable divergence in features learned by heterogeneous models, specialized information filtering method is essential to align features. We aim to emphasize the importance of noting that directly mimicking the feature space of a heterogeneous teacher can lead to suboptimal results. Instead, we find it more effective to transfer mismatched representations into the aligned logits space through the integration of additional exit branches within the student model. We leave the exploration of more efficient modules or alternative spaces for aligning features for future research.
>
> In essence, intermediate logits supervision can be viewed as a specialized approach for aligning heterogeneous features. Furthermore, we introduce a novel distillation loss to enhance target information adaptively by introducing a modulating parameter into the original KD loss. Our ablation study validates the efficacy of this design. We will provide a summary of these architecture designs in the related work section, including BYOT/DCM/DKS.
>
> **Response to the weakness 3:** Thanks for your valuable comments. Firstly, we would like to clarify the scope of our usage of the term "heterogeneous." In our paper, we specifically consider CNN, ViT, and MLP models as examples of heterogeneous architectures. However, in the context of SemCKD, the definition of heterogeneous models is broader. For instance, SemCKD categorizes models like VGG and ResNet as heterogeneous, whereas our paper treats them as homogeneous CNN models.
>
> We have also carried out experiments using SemCKD. Given that SemCKD is tailored for CNN models, we transformed intermediate features of ViT models into the CNN format using an "unpatchify" operation. Our experiments involve two teacher-student pairs: DeiT-T - ResNet18 and ConvNeXt-T - DeiT-T. To facilitate comparison, we present the results of SemCKD alongside KD and OFA on ImageNet-1K.
>
> |Teacher|Student|KD|SemCKD|OFA
> |:-:|:-:|:-:|:-:|:-:
> |DeiT-T|ResNet18|70.22|70.12|71.34
> |ConvNeXt-T|DeiT-T|74.00|71.96|74.41
>
> Through the above outcomes, within the more stringent "heterogeneous" context, SemCKD demonstrates lower performance compared to our OFA. We conjecture that the coarse "unpatchify" operation hinders effective information transfer. If we intend to extend the applicability of SemCKD to encompass generalized cross-architecture KD, a more meticulous design is imperative to achieve semantic calibration. We believe this as an important avenue for future research and plan to incorporate the discussion in our final revision.
>
> **Response to the weakness 4:** We employed CKA to illustrate that heterogeneous architectures learn distinct features, as indicated by their CKA similarity. This served as the foundation for devising feature alignment methods in the context of cross-architecture KD, enhancing the clarity and logical flow of the paper. While some existing works delve into different architectures, very few have simultaneously compared all three types. Hence, our analysis is more comprehensive. Furthermore, we haven't listed the CKA analysis as a specific contribution in our paper.
>
> **Response to the weakness 5:** Thanks for the valuable suggestions. Considering rebuttal time constraints, we restricted our experimentation to the ResNet50 teacher and MobileNet-v1/VGG-16 student pair. The particular ResNet-MobileNet teacher-student combination has been extensively utilized in numerous related studies, making it convenient to obtain baseline results. And we conducted the ResNet-VGG experiments ourselves to establish the baseline results. Following the recipe in DKD, we trained the student using our OFA and present the results in the following table.
>
> |Teacher|Student|KD|OFD|Review|CRD|DKD|DIST|OFA|
> |:-:|:-:|:-:|:-:|:-:|:-:|:-:|:-:|:-:|
> |ResNet50|MobileNet-v1|70.68|71.25|72.56|71.37|72.05|73.24|73.28|
> |ResNet50|VGG-16|73.96|73.31|74.08|74.02|74.69|74.75|74.88|
>
> From the results, the performance of OFA is comparable to the best baseline DIST with a slight accuracy gain of 0.04%. Notably, models mentioned above are considered homogeneous in our paper, as they all belong to the CNN architecture. Our OFA method is primarily tailored for cross-architecture KD and consistently surpasses previous baseline approaches. However, even when both the teacher and student belong to homogeneous architectures, OFA still manages to achieve competitive performance, underscoring its effectiveness.
>
> ### ***The part (2/2) can be found in "Author Rebuttal" at the top of this page.***

---

> > ### Comment · Reviewer_R7Hg · 2023-08-14
> > **Response**
> >
> > Thank you for your response. The rebuttal address most of my concerns. I have raised my score. I would be very glad to see the author release their code if this paper is accepted.

---

> > > ### Author Response · Authors · 2023-08-14
> > > **Response to Reviewer R7Hg**
> > >
> > > Dear Reviewer R7Hg,
> > >
> > > We sincerely appreciate you taking the time to review our paper and response, and contributing to improve this paper. We will carefully follow reviewer's advice to incorporate all the addressed points in the updated version. And we will release the code if our paper is accepted.
> > >
> > > Best,
> > > Authors

---

### Official Review · Reviewer_d3Lq · 2023-07-06

**Soundness:** 3 good
**Presentation:** 4 excellent
**Contribution:** 3 good
**Rating:** 7
**Confidence:** 5

**Summary:**

This paper tackles the problem of cross-architecture distillation, that is, the teacher and the student in KD are of different model architectures.

By using centered kernel alignment, the authors observe that features learned by models of different architectures shows significant feature divergence, indicating that previous hint-based methods are not suit for this task.

To bridge this gap, the authors propose a simple yet effective one-for-all KD framework called OFA-KD. Specifically, they project intermediate features into an aligned latent space to discard architecture-specific information.

And an adaptive target enhancement scheme is proposed to prevent the student from being disturbed by irrelevant information.

The authors conduct experiments on CIFAR-100 and ImageNet-1k benchmarks with CNN, ViT and MLP architectures to demonstrate the effectiveness of the proposed method.

**Strengths:**

Motivation: The motivation is clear. Cross-architecture distillation provides more feasible options for teacher models, as it may not always be possible to find a superior teacher model with a homogeneous architecture.

Originality: Learning in an aligned latent space is the first application in cross-architecture distillation, and the adaptive target information enhancement loss is novel.

Quality: Written of the paper is good. Sufficient experiments and ablation studies with other methods and the proposed variants are implemented.

Clarity: The paper consists of text explanations and an illustration of the OFA-KD framework and the proposed loss.

Significance: The paper solves the problem of cross-architecture distillation, expanding feasible options for teacher models in practice, and improves the accuracy of distilled student models.

**Weaknesses:**

1. In the CKA analysis, it seems that when comparing features of models of the same architecture, the authors just using features of one model in both x-axis and y-axis, as the corresponding heatmaps are symmetry. If using two models, such as ResNet18 vs. ResNet34 or two ResNet18 trained with different initialization, would the results be different?

2. The authors propose to using an aligned latent space for cross-architecture distillation, and adopting the logits space as a special instance. I wonder whether there are any other possible choices of the space, and what will happen when using them.

3. Introduction of the adaptive target information enhancement loss is a bit complicated. Maybe it is possible to simplify the notations.

4. What is the principle of designing the branches, such as its architecture and layer number? Have the authors tried other branch designs?

Additionally, there are some grammar mistakes and typos:

- line 192, "slow" -> "slowly"
- line 193, "enhance" -> "enhancement"
- line 255, "additional" -> "addition"
- line 257, "reports" -> "report"
- line 299, "strengthen" -> "strength"
- line 311, "methods" -> "method"
- line 339, "disitllation" -> "distillation"

**Questions:**

Please refer to "Weaknesses"

**Limitations:**

Yes. Limitations and border impact are discussed in the conclusion.

---

> ### Author Rebuttal · Authors · 2023-08-09
>
> > **Weakness 1:** In the CKA analysis, it seems that when comparing features of models of the same architecture, the authors just using features of one model in both x-axis and y-axis, as the corresponding heatmaps are symmetry. If using two models, such as ResNet18 vs. ResNet34 or two ResNet18 trained with different initialization, would the results be different?
>
> **Response to the weakness 1:** Thank you for your valuable suggestion. We have expanded our CKA analysis to encompass various model architectures, and presented these additional results in the PDF uploaded in "Author Rebuttal" section. We will also incorporate these results into our supplementary material. To conduct CKA analysis, specifically, we use ResNet18 and ResNet34 as CNN models, Swin-Small and Swin-Base as ViT models, Mixer-B/16 and ResMLP-S12 as MLP models. The results reveal that homogeneous models, such as a ResNet18 model and a ResNet34 model, exhibit similar feature learning at comparable positions within the model. Conversely, heterogeneous models continue to manifest distinct feature learning patterns. This corroborates the conclusions drawn in our main paper.
>
> > **Weakness 2:** The authors propose to using an aligned latent space for cross-architecture distillation, and adopting the logits space as a special instance. I wonder whether there are any other possible choices of the space, and what will happen when using them.
>
> **Response to the weakness 2:** The utilization of an aligned latent space aims to mitigate the adverse effects of disparate information present in the features learned by divergent teacher and student models. Therefore, the guiding principle in selecting an aligned latent space is to retain shared information while discarding irrelevant details. The logits space satisfies these prerequisites and is straightforward to implement, thus we have adopted it as the aligned latent space in our experiments.
>
> While we acknowledge that there might exist more efficient latent spaces for cross-architecture KD. For example, the use of manifold space as the latent space [1], which exclusively compares the relationships among learned features while remaining insensitive to the absolute feature values. However, crafting an optimal latent space remains a challenging task, given the absence of well-defined principles for measurement. Consequently, we consider this issue a subject for future research exploration.
>
> [1] Hao, Zhiwei, et al. "Learning efficient vision transformers via fine-grained manifold distillation." Advances in Neural Information Processing Systems 35 (2022): 9164-9175.
>
> > **Weakness 3:** Introduction of the adaptive target information enhancement loss is a bit complicated. Maybe it is possible to simplify the notations.
>
> **Response to the weakness 3:** Thank you for your suggestion. We have simplified the notations and incorporated them into our next version.
>
> > **Weakness 4:** What is the principle of designing the branches, such as its architecture and layer number? Have the authors tried other branch designs?
>
> **Response to the weakness 4:** We initially outline our approach to determining the block number within each branch. In our experimental setup, we partition all models into four stages. For pyramid architectures, this division is straightforward (four down-sampling layers). However, for models like DeiT, we adopt the division scheme used in the Swin Transformer architecture. For instance, DeiT-T and Swin-T both consist of 12 blocks. Thus, we divide DeiT-T into four stages, each comprising 2, 2, 6, and 2 blocks, mirroring the structure of Swin-T.
>
> Regarding the branch architecture, we adopt depth-wise convolutional blocks for CNN models and vision transformer blocks for ViT and MLP models. This design is rooted in our belief that a mismatch between branch and backbone architectures could hinder student performance. To verify this assumption, we conducted experiments comparing various branch architectures. The results, as shown in the table below, underscore that homogeneous pairings of branch and backbone outperform heterogeneous pairings. Additionally, our supplementary material presents PyTorch-style pseudocode for constructing branches.
>
> |  Teacher   | Student  | Branch architecture | Top-1 |
> | :--------: | :------: | :-----------------: | :---: |
> |   DeiT-T   | ResNet18 |         CNN         | 71.34 |
> |   DeiT-T   | ResNet18 |         ViT         | 70.82 |
> | ConvNeXt-T |  DeiT-T  |         CNN         | 74.27 |
> | ConvNeXt-T |  DeiT-T  |         ViT         | 74.41 |
>
> > **Weakness 5:** Additionally, there are some grammar mistakes and typos:
>
> **Response to the weakness 5:** Thank you for thoroughly reviewing our submission and pointing out the mistakes. We have taken great care to address these issues in our next version.

---

### Official Review · Reviewer_sLfN · 2023-07-07

**Soundness:** 3 good
**Presentation:** 3 good
**Contribution:** 3 good
**Rating:** 7
**Confidence:** 4

**Summary:**

This paper first demonstrates that there is significant feature divergence of the learned features between heterogeneous teacher and student models, which is a scenario rarely explored in previous knowledge distillation community. And the authors point out that the hint-based methods are ineffective in this cross-architecture distillation. They propose OFA-KD to improve the distillation performance between heterogeneous architectures. It first projects intermediate features into an aligned latent space (the logits space). In addition, they introduce an adaptive target enhancement scheme to prevent the student from being disturbed by irrelevant information. The extensive experiments with various architectures demonstrate the superiority of the OFA-KD framework.

**Strengths:**

This paper studies KD with different architectures. It is an interesting attempt to build a generic framework for distilling students with arbitrary mainstream model architectures, i.e., CNN, ViT, and MLP. Experiment results in both the main paper and the supplementary material demonstrate the necessity of doing cross-architecture distillation.

The performance improvement of OFA is remarkable. On ImageNet-1K, the maximum improvement is 0.8\%, and on CIFAR-100, the maximum improvement is 5.0\%, even compared with the most recent KD baselines, DIST and DKD.


**Weaknesses:**

The additional branches will increase the training cost. I think the authors should give more analyses on this.

The reported results of using res50 as the teacher on ImageNet-1K are obtained by using both FitNet and OFA-KD (Table 1). I think this is not a fair comparison. For example, what’s the result of FitNet + DKD/DIST?

Missing some references also adopting multi-branch architecture:
[A] Be your own teacher: Improve the performance of convolutional neural networks via self distillation, ICCV 2019
[B] Distillation-based training for multi-exit architectures, ICCV 2019
[C] MSD: Multi-Self-Distillation Learning via Multi-classifiers within Deep Neural Networks, arXiv 2019

**Questions:**

The OFA result (ViT-B teacher) in Table 6 of the main paper seems inconsistent with that in Figure 5 of the supplementary material. The accuracy gain of the res50 student is 1.19\% in the main paper, while in the supplementary material, the accuracy gain is 1.47\%. Please check it.

As there are additional branches introduced during the training procedure, it requires more computational resources to train the same student model than using traditional KD methods. Could the authors provide more details about the branches to illustrate the extra resources consumption? For example, I am interested in the number parameters and the FLOPs of the additional branches.

Why does the paper choose to use depth-width convolutional layers for branches in CNN models, while ViT blocks are used in ViTs and MLPs?

There are so many hyperparameters such as the scaling factor, the clip grad norm, and the \gamma value. How to choose these hyperparameters in practice?

**Limitations:**

Yes. Limitations and border impact are discussed in the conclusion.

---

> ### Author Rebuttal · Authors · 2023-08-09
>
> > **Weakness 1:** The additional branches will increase the training cost.
> >
> > **Question 2:** Could the authors provide more details about the branches
>
> **Response to the weakness 1 & question 2:** Thanks for your valuable comments. The inclusion of extra branches inevitably demands more computational resources during training. To mitigate this concern, we have employed a streamlined design for branch architecture. For example, in CNN models, we've reduced the number of channels in depth-wise convolution layers within the branches, and for ViT and MLP models, we've introduced patch merging blocks before vision transformer blocks within the branches to reduce the number of tokens that need to be processed.
>
> We've chosen three distinct student architectures, i.e., ResNet18, DeiT-T, and ResMLP-S12, and conducted a comparison on the number of parameters and FLOPs between their backbones and exit branches. As depicted in the table below, considering the additional FLOPs introduced by exit branches, these branches demonstrate enhanced accuracy improvement in comparison to the backbone architectures. This improvement results in only a marginal increase in training cost, thereby contributing minimal augmentation to overall training expenses. Moreover, given that the primary objective of KD is to compress a pre-trained model for easier deployment, a slight elevation in training cost remains acceptable. Our OFA-KD approach enhances the performance of student models while incurring no additional inference cost.
>
> | Teacher  |  Student   | Student Params | Student FLOPs | Branch Params | Branch FLOPs |
> | :------: | :--------: | :------------: | :-----------: | :-----------: | :----------: |
> |  DeiT-T  |  ResNet18  |     11.69M     |     1.83G     |     3.04M     |    0.08G     |
> | ResNet50 |   DeiT-T   |     5.72M      |     1.26G     |     4.48M     |    0.19G     |
> | ResNet50 | ResMLP-S12 |     15.35M     |     3.01G     |    16.32M     |    0.74G     |
>
> > **Weakness 2:** result of FitNet + DKD/DIST?
>
> **Response to the weakness 2:** FitNet appears to be particularly effective as a knowledge distillation approach when utilizing ResNet50 as the teacher and ViT/MLP as the student. However, these promising outcomes are not observed with other model combinations. As a result, we exclusively present results obtained by combining OFA and FitNet for these specific teacher-student pairs. To ensure a fair comparison, we proceed to conduct experiments merging FitNet with DKD and DIST for these models, and subsequently report their respective performances in the table below.
>
> | Teacher  |  Student   | DKD+FitNet | DIST+FitNet | OFA+FitNet |
> | :------: | :--------: | :--------: | :---------: | :--------: |
> | ResNet50 |   DeiT-T   |   75.60    |    75.13    |   76.55    |
> | ResNet50 |   Swin-N   |   78.23    |    77.95    |   78.64    |
> | ResNet50 | ResMLP-S12 |   78.23    |    77.71    |   78.53    |
>
> As the results demonstrate, even though DKD and DIST also yield improved performance when integrated with FitNet, our OFA-KD method consistently outperforms them. This underscores the efficacy of our approach. We will incorporate the new results obtained through FitNet + DKD/DIST into our next version."
>
> > **Weakness 3:** Missing some references
>
> **Response to the weakness 3:** Thank you for your valuable suggestion. We have appropriately integrated these references into the draft of our next version.
>
> > **Question 1:** the inconsistent result.
>
> **Response to the question 1:** Thank you for your thorough review of our paper. There was an error in our main paper regarding the accuracy gain, which should be 1.47%. We have rectified this mistake in the revised version.
>
> > **Question 3:** principle of choosing branches
>
> **Response to the question 3:** Our decision to adopt the CNN branch architecture for CNN models and the ViT branch architecture for ViT/MLP models is grounded in our belief that a heterogeneous pairing of branches and backbones could potentially hinder student performance. In order to substantiate this perspective, we conduct a series of experiments aimed at evaluating the outcomes of employing diverse branch and backbone configurations.
>
> The results below show that a homogenous combination of branch and backbone consistently yielded superior performance. As a result, we embraced this unified configuration, aligning with our objective of improving student model performance. Furthermore, the incorporation of depth-wise convolutional layers within the CNN branch architecture serves to mitigate the additional training cost attributed to branches. These convolutional layers have also demonstrated their effectiveness in several related studies [1,2].
>
> |  Teacher   | Student  | Branch architecture | Top-1 |
> | :--------: | :------: | :-----------------: | :---: |
> |   DeiT-T   | ResNet18 |CNN| 71.34 |
> |   DeiT-T   | ResNet18 |ViT| 70.82 |
> | ConvNeXt-T |  DeiT-T  |CNN| 74.27 |
> | ConvNeXt-T |  DeiT-T  |ViT| 74.41 |
>
> [1] Be your own teacher: Improve the performance of convolutional neural networks via self distillation
> [2] Task-oriented feature distillation
>
> > **Question 4:** How to choose these hyperparameters
>
> **Response to the question 4:** In our ablation study, we evaluated the impact of these hyperparameters. Notably, we discovered that the scaling factor can be disregarded by simply setting it to 1, as this adjustment is sufficient to achieve satisfactory results. As for the clip grad norm, the best result is obtained when the value is set to 5, which is a common setting used by many other related works. So we can simply adopt the setting of clip_grad=5 for all combinations of teacher and student. While there isn't a universally applicable setting for the parameter $\gamma$, we can determine its optimal value using a validation set, considering that the other hyperparameters remain fixed.

---

### Author Rebuttal · Authors · 2023-08-09

# Response to all reviewers

We thank all the reviewers for their elaborate and constructive feedback. Their valuable suggestions help improve the quality of our paper greatly.

### **Response to Reviewer R7Hg part (2/2)**

**Response to the question 1:** In our experiments, we primarily utilize the training script from the timm library to train the student models, while also implementing the KD loss of the baselines following the DKD approach. However, due to the distinct nature of features learned by heterogeneous architectures, certain differences arise. For example, the feature shape of a CNN is of size (N, C, H, W), while that of a ViT is denoted as (N, L, D), where N indicates the batch size, C, H, and W refer to the channel, height, and width of the CNN model's feature map respectively, and L and D denote the patch number and embedding dimension of the ViT/MLP model's feature map. To apply previous feature distillation methods designed for CNN models, we need to transform the feature map of the ViT/MLP model into the CNN-style (shape) feature through an "unpatchify" operation. However, we speculate that this operation might be overly simplistic and overlooks intrinsic features specific to the ViT/MLP model's learned features. Consequently, some feature distillation methods yield suboptimal results compared to the baseline. Furthermore, we believe that more effective approaches can be developed to adapt existing feature distillation methods to cross-architecture KD, but we leave this question for future research.

Generally speaking, the performances from logits-based algorithms are superior to those from hint-based algorithms. This observation reinforces our conclusion that for heterogeneous architectures, directly learning the intermediate features of the teacher might lead to sub-optimal results. However, there are a few instances where logits-based methods fall short of the baseline (student trained from scratch) performance. This discrepancy could potentially be attributed to the distinct inductive biases of CNN and ViT architectures, which drive them toward diverse destinations and result in dissimilar distributions. As discussed in Lines 173-175 of our main text, this phenomenon motivates us to propose the adaptive target information enhancement KD loss.

**Response to the question 2:** We mainly adopt the top-1 accuracy as the measurement to evaluate different methods in our paper. Existing methods cannot achieve consistent top-1 accuracy improvement across all tested combinations of teacher and student. However, our OFA-KD outperforms those baselines consistently in terms of top-1 accuracy. This is an evidence that demonstrating OFA-KD is a generic method for cross-architecture distillation.

In this paper, we predominantly employ top-1 accuracy as the evaluation metric to assess various methods. The previous approaches do not uniformly achieve improvements in top-1 accuracy across all tested teacher-student combinations. Conversely, our OFA-KD consistently outperforms these baselines in terms of top-1 accuracy. This consistent superiority stands as evidence that underscores OFA-KD's effectiveness as a method for cross-architecture distillation. While we haven't identified another quantitative indicator at present, this area presents an intriguing avenue. Exploring more comprehensive ways to assess the performance of KD methods is a topic we view as a potential focus for future research.

---

### Decision · Program_Chairs · 2023-09-21

**Decision:**

Accept (poster)

**Comment:**

This work first demonstrated the existence of feature divergence between heterogeneous teacher and student models, which is rarely discussed in previous knowledge distillation methods. Based on this observation, the authors propose a one-for-all KD framework (OFA-KD) where the intermediate features will be projected into an aligned latent space to discard architecture-specific information, and an adaptive target enhancement scheme is proposed to prevent the student from being disturbed by irrelevant information. Despite the additional training costs and the necessary need for hyper-parameter tuning, most reviewers agreed with the contribution of this paper to the community.